# MATS: An Audio Language Model under Text-only Supervision

Wen Wang[1]  Ruibing Hou[1 †]  Hong Chang[1 2]  Shiguang Shan[1 2]  Xilin Chen[1 2]

## Abstract

Large audio-language models (LALMs), built upon powerful Large Language Models (LLMs), have exhibited remarkable audio comprehension and reasoning capabilities. However, the training of LALMs demands a large corpus of audio-language pairs, which requires substantial costs in both data collection and training resources. In this paper, we propose **MATS**, an audio-language multimodal LLM designed to handle **M**ultiple **A**udio task using solely **T**ext-only **S**upervision. By leveraging pre-trained audio-language alignment models such as CLAP, we develop a text-only training strategy that projects the shared audio-language latent space into LLM latent space, endowing the LLM with audio comprehension capabilities without relying on audio data during training. To further bridge the modality gap between audio and language embeddings within CLAP, we propose the **S**trongly-related **n**oisy **t**ext with **a**udio (**Santa**) mechanism. Santa maps audio embeddings into CLAP language embedding space while preserving essential information from the audio input. Extensive experiments demonstrate that MATS, despite being trained exclusively on text data, achieves competitive performance compared to recent LALMs trained on large-scale audio-language pairs. The code is publicly available in https://github.com/wangwenbanban/MATS.

## 1. Introduction

Recent advancements in multimodal models for audio processing have progressed rapidly, driven by the recognition

[1]Key Laboratory of Intelligent Information Processing of Chinese Academy of Sciences (CAS), Institute of Computing Technology, CAS, China [2]University of Chinese Academy of Sciences, China. Correspondence to: Ruibing Hou <houruibing@ict.ac.cn>.

*Proceedings of the 42nd International Conference on Machine Learning*, Vancouver, Canada. PMLR 267, 2025. Copyright 2025 by the author(s).

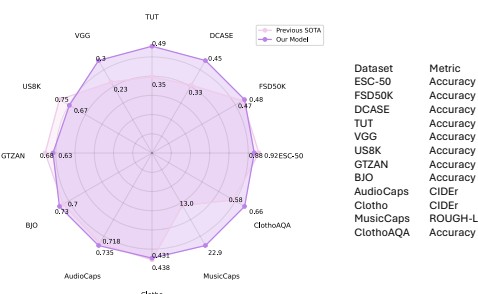

*Figure 1.* Performance of MATS compared to the previous SOTA (Chu et al., 2023; Kong et al., 2024; Deshmukh et al., 2023) under zero-shot setting for close-ended audio tasks.

of audio as a crucial role in understanding the physical world. Audio, encompassing sound, music, and other auditory elements, plays a crucial role in enabling intelligent agents to assist humans and navigate the complexities of the real world. Consequently, developing models capable of effectively interpreting these audio modalities is vital for advancing human-machine interaction and fostering a deeper understanding of our surroundings.

In the field of audio-language multimodal learning, contrastive learning models, such as Contrastive Language-Audio Pretraining (CLAP) (Elizalde et al., 2023), have demonstrated remarkable zero-shot abilities across various audio tasks. However, these models primarily specialize in audio discriminative tasks, lacking a decoder to support open-ended audio question-answering (QA) capabilities. To overcome this limitation, recent studies have integrated LLMs as the text generator within a multimodal framework, capitalizing on LLMs' inherent instruction following and generating capabilities. A pioneering effort, Pengi (Deshmukh et al., 2023), integrates CLAP audio encoder and GPT2 (Radford et al., 2019), employing a transformer projection module to align audio embedding with LLMs' text embedding space. Building on this foundation, recent models (Gong et al., 2024; Tang et al., 2024; Kong et al., 2024; Ghosh et al., 2024) utilize advanced audio encoder and more powerful LLMs, constructing advanced LALMs to address increasingly complex audio relevant tasks.

However, training a large audio-language model that effectively bridges the entirely decoupled audio embedding space and LLM text embedding space is challenging. Previous models (Gong et al., 2024; Tang et al., 2024; Kong et al., 2024; Ghosh et al., 2024) rely heavily on extensive audio-language QA pairs for training. This requires annotators to carefully listen to each audio and then construct corresponding questions and answers, a labor-intensive and time-consuming process. As a result, the cost of data collection becomes prohibitively high, leading to significant resource demands for training a LALM.

To reduce resource demands, researchers have proposed text-only audio captioning frameworks (Deshmukh et al., 2024; Kouzelis & Katsouros, 2023; Zhang et al., 2024; Li et al., 2024) which generate audio captions without relying on costly audio-language pairs for training. These frameworks leverage CLAP, which have established a well-aligned audio-language embedding space, enabling LLMs to interpret audio embedding by linking CLAP language embedding space with LLM semantic space. However, these methods have certain limitations. First, they are narrowly tailored to specific audio type (sound) and constrained to single audio captioning task, limiting their applicability to boarder audio open-ended question-answering scenarios. Second, to bridge the audio-language modality gap within CLAP, these methods either rely solely on noise injection during training or exclusively utilize a memory bank during inference. However, using only noise injection is highly random and sensitive, while relying solely on a memory bank during inference not only sacrifices audio embedding information but also increases inference time, ultimately compromising overall effectiveness.

In this paper, we propose **MATS**, an audio-language multimodal LLM designed to address **M**ultiple **A**udio tasks, encompassing both close-ended and open-ended scenarios, using solely **T**ext-only **S**upervision. MATS builds upon recent advancements in LALMs (Deshmukh et al., 2023; Tang et al., 2024), incorporating two key enhancements. **First**, recognizing that the joint audio-language embedding space of a pre-trained CLAP is shared across both modalities, the language embeddings extracted by CLAP language encoder can serve as effective representations of audio semantics. During training, MATS integrates CLAP language encoder, a Mapper and a LLM, where the CLAP language encoder extracts cross-modality embeddings of audio descriptions within the shared language-audio embedding space. During inference, the LLM can generate responses based on audio embedding derived from input audios, processed through CLAP audio encoder and the Mapper. **Second**, we derive the generalization error bound for text-only supervised multimodal models, revealing that the inherent modality gap in contrastive audio-language representation learning can amplify the generalization error bound. To this end, we

propose a modal-transfer method that integrates noise injection during training and employs **S**trongly-rel**a**ted **n**oisy **t**ext with **a**udio (**Santa**) during inference. And the Santa mechanism employs a k-means-based memory alongside a balancing strategy. This design effectively balances audio embeddings and semantically strongly related augmented language embeddings, effectively mitigating the modality gap and enhancing MATS's generalization ability.

We conduct a comprehensive evaluation of our model on both close-ended and open-ended tasks. Remarkably, despite being trained on text-only data, MATS demonstrates performance comparable to, and oven surpassing, some LALMs trained on extensive audio-language pairs. For close-ended tasks, MATS demonstrates significant zero-shot performance improvements over previous state-of-the-arts (SOTAs) (Deshmukh et al., 2023; Tang et al., 2024; Doh et al., 2023), achieving gains of **12%**, **7%**, and **9.9%** on DCASE (Mesaros et al., 2017), VGG (Chen et al., 2020), and MusicCaps (Agostinelli et al., 2023) benchmarks, respectively, as shown Figure 1. For open-ended tasks, MATS (7B) achieves the second-best performance on MMAU benchmark (Sakshi et al., 2024), surpassing SALMONN (13B) (Tang et al., 2024) by **11.7%** with a significantly smaller model size.

## 2. Related work

**Audio Language Models.** In the field of audio-language models, CLAP models (Wu et al., 2023; Elizalde et al., 2023) have demonstrated remarkable capabilities in audio discriminative tasks. However, the lack of a decoder limits their applicability in open-ended QA scenarios. Recently, with the rapid advancements in LLMs, researchers have started integrating audio understanding into LLMs. For example, Pengi (Deshmukh et al., 2023) combines the CLAP audio encoder with GPT2 (Radford et al., 2019), and employs a transformer-based mapper for multimodal fusion, achieving strong performance on close-ended tasks. Similarly, LTU (Gong et al., 2024) incorporates a more advanced LLM, LLaMA (Touvron et al., 2023), and demonstrates emerging audio comprehension and reasoning abilities. SALMONN (Tang et al., 2024) utilizes a dual audio encoder consisting of a Whisper speech encoder model (Radford et al., 2023) and a BEATs (Chen et al., 2023) audio encoder to handle speech and non-speech audio tasks effectively. Other studies have expanded instruction fine-tuning data and explored more advanced audio encoder architecture, such as GAMA (Ghosh et al., 2024), Qwen2-Audio (Chu et al., 2024), yielding commendable performance.

However, these LALMs heavily rely on large-scale audio-language pairs for training, which pose significant challenges in term of data collection and training costs. Differently, we adopt a highly cost-efficient approach for training

LALMs using text-only data, which substantially reduces the data collection and training overhead while maintaining comparable performance.

**Text-only Supervised Multimodal LLMs.** To alleviate resource demands, researchers have proposed zero-shot captioning frameworks, aimed at generating image/audio captions through text-only training. In the visual domain, large-scale pre-trained contrastive models like CLIP (Radford et al., 2021) align images and language into a shared vision-language embedding space. Building on CLIP, CapDec (Nukrai et al., 2022) trains a decoder to reconstruct text from its corresponding CLIP language embedding, which is then used to decode CLIP image embeddings at inference. To mitigate the vision-language modality gap, CapDec injects noise into language embedding during training. Differently, DeCap (Li et al., 2023) leverages a memory to store CLIP language embeddings, which is subsequently used to project visual embedding into CLIP language embedding space at inference. In the audio domain, models such as NoAudio-Captioning (Deshmukh et al., 2024), WSAC (Kouzelis & Katsouros, 2023), PromptAAC (Zhang et al., 2024), and DRCap (Li et al., 2024) adopt similar strategies, replacing CLIP with CLAP to develop zero-shot audio captioning frameworks. However, these works focus on audio captioning, lacking the flexibility to handle a diverse range of audio tasks simultaneously, and they are limited to processing only a single type of audio (sound).

## 3. Methodology

In this section, we introduce **MATS**, an audio-language multimodal LLM designed to tackle multiple audio tasks using text-only supervision. We begin by presenting the formulation of MATS in Section 3.1, followed by an outline of its overall framework in Section 3.2. Section 3.3 provides the theoretical foundation, demonstrating that reducing the feature distribution gap between the CLAP's audio and language modalities can effectively reduce the generalization error of MATS. Section 3.4 presents the proposed **Santa** mechanism to bridge the audio-language modality gap. Finally, the training pipeline is detailed in Section 3.5.

### 3.1. Formulations

As shown in Figure 2, **MATS** consists of a language encoder $\mathcal{E}_T$ and an audio encoder $\mathcal{E}_A$ from CLAP (Elizalde et al., 2023). During training, the language encoder $\mathcal{E}_T$ processes text data $T$, while during inference, the audio encoder $\mathcal{E}_A$ processes audio data $A$. To bridge the audio-language modality gap, the noise injection is used during training and the Santa mechanism $f_{\text{Santa}}$ is employed at inference. Then, a transformer-based mapper $f_{\text{map}}$ serves as the connection module, integrating its output sequence with the text instruction prompt, which is then fed into the LLM to generate the

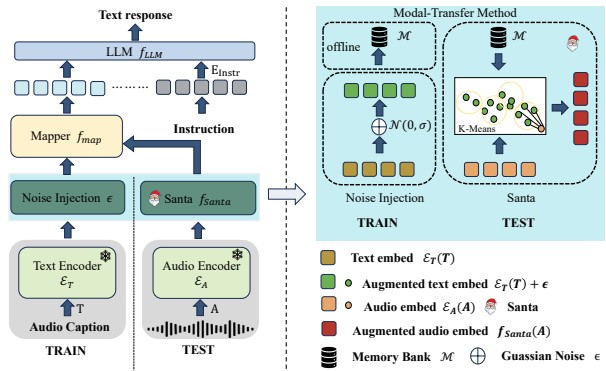

*Figure 2.* The architecture of proposed MATS.

text response.

**Train Phase.** During training, only text data is utilized. Formally, the input text $T$, which provides the descriptions of audio contents, is encoded by CLAP language encoder $\mathcal{E}_T$. To mitigate the audio-language modality gap, a zero-mean Gaussian noise is added to the extracted language embedding $\mathcal{E}_T(T)$. The augmented language embedding is subsequently projected through the mapper $f_{\text{map}}$ to generate embeddings within the LLM latent space. These embeddings are concatenated with the instruction embeddings $E_{\text{Instr}}$ extracted by LLM's word embedding layer, forming the prefix $s_{\text{train}}$, which serves as the input to the LLM for autoregressive learning. The process is formulated as:

$$s_{\text{train}} = \text{concat}\left[f_{\text{map}}\left(\mathcal{E}_T\left(T\right) + \epsilon\right), E_{\text{Instr}}\right], \quad (1)$$

where noise $\epsilon \sim \mathcal{N}(0, \sigma)$ with variance $\sigma$. Given the prefix $s_{\text{train}}$ and corresponding text response $R$, the LLM predicts the probability distribution of potential next token at each step, with the Cross-Entropy loss function used for optimization:

$$\mathcal{L} = \sum_i \text{CE}\left(f_{\text{LLM}}\left(s_{\text{train}}, R^{<i}\right), R^i\right), \quad (2)$$

where CE denotes the cross-entropy function.

**Inference Phase.** At inference, the input data includes both audio files and textual instructions. The key structure difference is that the augmented language embedding $\mathcal{E}_T(T) + \epsilon$ is replaced by audio embedding $\mathcal{E}_A(A)$, encoded by the audio encoder of CLAP. Notably, the **Santa** mechanism is then applied to the audio embedding to mitigate the audio-language modality gap. Finally, the LLM predicts the responses $R_{\text{pred}}$ in an autoregressive manner, as:

$$s_{\text{test}} = \text{concat}\left[f_{\text{map}}\left(f_{\text{Santa}}\left(\mathcal{E}_A\left(A\right)\right)\right), E_{\text{Instr}}\right],$$
$$R_{\text{pred}} = f_{\text{LLM}}\left(s_{\text{test}}\right). \quad (3)$$

### 3.2. Model Architecture

The architecture of the MATS model is depicted in Figure 2.

**CLAP Encoder.** To obtain an aligned audio-language embedding space, we adopt CLAP (Elizalde et al., 2023), which integrates the HTSAT (Chen et al., 2022) audio encoder $\mathcal{E}_A$ and the GPT2 (Radford et al., 2019) language encoder $\mathcal{E}_T$. CLAP is trained to connect language and audio by corresponding encoder and bring them into a joint multimodal space using contrastive learning. Both encoders are kept frozen throughout the entire process. Notably, during training, the language encoder $\mathcal{E}_T$ is utilized, whereas the audio encoder $\mathcal{E}_A$ is employed during inference.

**Modality Transfer.** To mitigate the audio-language modality gap within CLAP and enhance the model's generalization, we add zero-mean Guassian noise to the language embedding during training as $\mathcal{E}_T(T) + \epsilon$, where $\epsilon \sim \mathcal{N}(0, \sigma)$. The variance $\sigma$ is determined by calculating the infinity norm between audio and language embeddings over a set of 30 randomly selected samples following (Deshmukh et al., 2024). During inference, the augmented language embedding $\mathcal{E}_T(T) + \epsilon$ is replaced by the audio embedding $\mathcal{E}_A(A)$, and the **Santa** mechanism is applied to the audio embedding to further reduce the audio-language modality gap within CLAP, formulating as $f_{\text{Santa}}(\mathcal{E}_A(A))$. The Santa mechanism is detailed in Section 3.4.

**Mapper Module.** The mapper module consists of a 8-layer Transformer (Vaswani, 2017) followed by a linear layer. Specifically, we introduce a fixed set of learnable query embeddings, which are concatenated with the embeddings produced by the CLAP encoder and then fed into the Transformer. Through the self-attention mechanism in transformer, these queries interact with the frozen CLAP embeddings, enabling effective integration of their information. Finally, the output query embeddings are projected via a linear layer to align with the dimensionality of the LLM's text embeddings.

**Large Language Model.** To generate textual responses, we utilize a pretrained autoregressive causal language model. In this study, we evaluate two large language models: the smaller version employs GPT2 with 125M parameter, while the larger version leverages the LLaMA-7B model fine-tuned with Vicuna instruction-following capabilities. These two LLMs are selected to balance computational efficiency with performance, enabling a comparative analysis of lightweight and advanced language generation capabilities.

### 3.3. Theoretical Analysis on Generalization

In this subsection, we study the generalization error bound of the text-only supervised audio models. Formally, let $\mathcal{A}$ and $\mathcal{T}$ be input audio and corresponding textual descriptions space respectively, and $\mathcal{Y}$ be output one-hot class space, where the one-hot vector $y \in \mathbb{R}^V$ is used to represent the label and $V$ denotes the LLM's vocabulary size. Similarly, $\mathcal{Z}^t$ stands for the embedding space induced from

$\mathcal{T}$ by the CLAP language encoder, and $\mathcal{Z}^a$ stands for the embedding space induced from $\mathcal{A}$ by the CLAP audio encoder. Additionally, let $h$ denote the prediction function mapping from the CLAP embedding space to output space, i.e., $h : \mathcal{Z}^t/\mathcal{Z}^a \to \mathcal{Y}$.

To simplify the analysis, we focus on a specific audio event classification task and disregard the influence of instructions. In this case, we are given a training set consisting of text-label paired examples, $\mathcal{D}_{tr} = \{z_i^t, y_i\}_{i=1}^N$ where $(z_i^t, y_i) \sim p_{\mathcal{T}}(z^t, y)$. Our goal is to learn a target model $h : \mathcal{Z}^a \to \mathcal{Y}$ by fitting on $\mathcal{D}_{tr}$, with the smallest generalization risk on the test distribution $p_{\mathcal{A}}(z^a, y)$ where $(z^a, y)$ denotes audio-label pairs. Formally, the empirical error on training set and the generalization risk on test set are computed as:

$$
\widehat{R}_{tr}(h) = \frac{1}{N} \sum_{i=1}^{N} \left| h\left(z_i^t\right) - y_i \right|,
$$
$$
R_{te}(h) = \mathbb{E}_{(z^a, y) \sim p_{\mathcal{A}}(z^a, y)} \left| h\left(z^a\right) - y \right|,
$$
(4)

where $|\cdot|$ is $L_1$ norm. With these definitions, we can derive the generalization risk bound of the text-only supervised audio model, leading to following theorem.

**Theorem 3.1.** *Let $\mathcal{H}$ be a hypothesis space of Natarajan-dimension $d$. For classification with $V$ classes, let $\mathcal{D}_{tr}$ be text-only training set drawn from distribution $p_{\mathcal{T}}(z^t, y)$. As for test set with audio files draw from distribution $p_{\mathcal{A}}(z^a, y)$, assuming that the class distribution of training and test data is consistent, for any target classifier $h \in \mathcal{H}$, $\delta \in (0, 1)$, with probability at least $1 - \delta$:*

$$
\begin{aligned}
R_{te}(h) \leq\ & \widehat{R}_{tr}(h) \\
& + \max_v \left( \text{disc}_{L_1} \left( p_{\mathcal{A}}\left(z^a | y\right), p_{\mathcal{T}}\left(z^t | y\right) \right) \right) \\
& + \sqrt{\frac{8}{N} \left( 2d \log \sqrt{2NV} + \log \frac{2}{\delta} \right)}.
\end{aligned}
$$
(5)

*Here, $\text{disc}_{L_1}$ denotes the Discrepancy Distance (Definition 4 in (Mansour et al., 2009)).*

The proof is provided in Appendix A. Theorem 3.1 reveals that the generalization risk can be bounded by the sum of three components: (1) *Empirical Risk:* $\widehat{R}_{tr}(h)$, representing the empirical risk on training set $\mathcal{D}_{tr}$, which can be reduced during training. (2) *Modality Discrepancy:* The second term denotes the discrepancy between CLAP audio embeddings and language embeddings. (3) *Complexity Term:* The last term is only relevant with the choice of the hypothesis space $\mathcal{H}$ and magnitude of $N$. In general, this term vanishes when the training data is sufficiently large. These observations highlight a crucial factor in reducing the generalization error bound: minimizing the audio-language modality gap within CLAP.

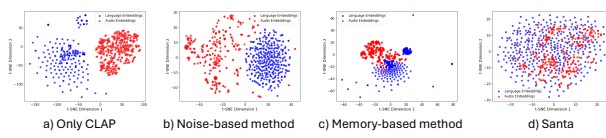

*Figure 3.* t-SNE visualizations for various methods on 350 randomly selected language embeddings (blue) and paired audio embeddings (red) from training dataset of MATS-Audio in Table S9. a) CLAP b) CLAP with Gaussian Noise (Deshmukh et al., 2024) c) CLAP with memory bank (Li et al., 2024) d) CLAP with Santa.

### 3.4. Santa Mechanism

According to theoretical analysis above, reducing *Modality Discrepancy* is critical for improving MATS's generalization. However, CLAP suffers from an inherent modality gap: As shown in Figure 3 a), CLAP's audio and language embeddings occupy entirely separate regions. To address this issue, (Deshmukh et al., 2024) injects noise into CLAP's language embeddings during training. However, relying solely on noise injection is highly stochastic and sensitive, limiting its robustness. Another work (Li et al., 2024) utilizes a memory bank that integrates similar language embeddings to represent audio embeddings during inference. However, this approach completely discard original audio embeddings, resulting in a significant loss of audio information. In this work, we propose modality transfer, which combines noise injection during training with the **Santa** mechanism during inference. The **Santa** mechanism utilizes a k-means-based memory alongside a balancing strategy. This design integrates audio embeddings with semantically relevant augmented language embeddings, more effectively bridging the modality gap, as shown in Figure 3 d).

To obtain the CLAP language embedding space augmented with noise injection, we randomly select $M$ texts from the text-only training set, denoted as $\{T_i\}_{i=1}^{M}$, and construct a memory $\mathcal{M} = \{\mathbf{m_i}\}_{i=1}^{M}$, where $\mathbf{m_i} = \mathcal{E}_T(T_i) + \epsilon$. At inference, the goal is to generate an answer for a given audio $A$ and associated instruction. Specifically, given the audio embedding $z^a = \mathcal{E}_A(A)$, we obtain its representation in language embedding space by combing $z^a$ with a weighted sum of semantically relevant embeddings in memory $\mathcal{M}$. A straightforward strategy to obtain the weights of these language embeddings is to compute their similarity with the audio embedding. However, due to the limited representational power of individual language embedding, this strategy is prone to retrieving the texts with insufficient semantic relevance, thereby affecting the effectiveness of audio-language modality alignment.

To this end, we apply k-means clustering to $\mathcal{M}$ and obtain a clustered memory $\mathcal{S} = \{\mathcal{S}_k\}_{k=1}^{K}$, where $\mathcal{S}_k$ stores the augmented language embeddings assigned to this cluster. By aggregating semantically similar augmented language embeddings, each cluster exhibits enhanced semantic rep-

resentational capabilities. Denote the cluster centers as $\{\mathbf{c_k}\}_{k=1}^{K}$, we identify the most relevant cluster $\mathcal{S}_{\text{closest}}$ by calculating the distance between the audio embedding $z^a$ and the cluster centers:

$$\text{closest} = \arg\min_{k} \|z^a - \mathbf{c_k}\|_2^2. \quad (6)$$

Next, we select $L$ augmented language embeddings from the cluster $\mathcal{S}_{\text{closest}}$, denoting as $\{\mathbf{m_1^{\text{closest}}}, \ldots, \mathbf{m_L^{\text{closest}}}\}$, with the top $L$ closest distances to the audio embedding. Finally, we compute a weighted sum of these $L$ augmented language embeddings and balance its contribution with the audio embedding. The weights of these language embeddings are derived by calculating the dot-product similarity between $z^a$ and each $\mathbf{m_i^{\text{closest}}}$, normalized via a softmax function. This overall **Santa** process is formalized as:

$$f_{\text{Santa}}(z^a) = (1 - \lambda) z^a + \lambda \sum_{i=1}^{L} w_i * \mathbf{m_i^{\text{closest}}},$$

$$\text{where, } w_i = \frac{\exp\left(\left((z^a)^T \mathbf{m_i^{\text{closest}}}\right)/\tau\right)}{\sum_{j=1}^{L} \exp\left(\left((z^a)^T \mathbf{m_j^{\text{closest}}}\right)/\tau\right)}, \quad (7)$$

where $\tau$ is a temperature hyperparameter and $\lambda$ is a balancing hyperparameter.

### 3.5. Training Pipeline

**AudioTIA-5M Dataset.** MATS is designed to handle a wide range of audio tasks relying solely on **text** supervision, which makes existing audio datasets unsuitable for direct use in its training. In this work, we construct a new dataset `AudioTIA-5M`, where each training sample is formatted as (`text`, `instruction`, `answer`) tuple. `AudioTIA-5M` encompasses both close-ended and open-ended tasks, employing four audio-task templates in Table S11 of Appendix. For the classification task (CLS), we leverage ChatGPT (Achiam et al., 2023) to generate a substantial amount of textual data. Specifically, by utilizing ChatGPT's instruction-following and text generation capabilities, we create a diverse set of audio descriptions (used as `text`) based on given event labels (used as `answer`). Further details are provided in Appendix B.4. For the audio captioning task (CAP), we directly use audio caption annotations from public datasets, including Clotho-v2 (Drossos et al., 2020), AudioCaps (Kim et al., 2019), MusicCaps (Agostinelli et al., 2023), WavCaps (Mei et al., 2024) and Macs (Martín-Morató & Mesaros, 2021), as `text` and `answer`. To further enhance model's understanding and reasoning capabilities for audio, we incorporate two open-ended datasets: OpenAQA (Gong et al., 2024) and MusicQA (Liu et al., 2024a). In OpenAQA, we replace raw audio files with their audio captions as model inputs. For MusicQA, where music captions are unavailable, we generate corresponding captions using LP-MusicCaps (Doh et al., 2023).

Overall, `AudioTIA-5M` comprises two subsets: a 1.5 M close-ended question subset, and a 3.8 M open-ended question subset. The detailed statistics of `AudioTIA-5M` are presented in Table S8 of Appendix.

**Training.** To maintain alignment between language and audio embeddings within the CLAP model while leveraging the extensive knowledge and generative capacities of LLM, we freeze both CLAP and LLM model, training only the mapper module and LLM's LoRA adapters (Hu et al., 2022). Our model is trained on `AudioTIA-5M` dataset, which integrates both close-ended and open-ended tasks. The close-ended tasks equip the model with foundational audio perception capabilities, while the open-ended tasks enhance its advanced reasoning abilities. Training is conducted in an autoregressive manner using Equation 2.

# 4. Experiment

In this section, we conduct extensive experiments on both close-ended and open-ended tasks. The details about training and test benchmarks are provided in Appendix B. Additional extensive hyperparameter analysis and visualizations are provided in Appendix C.

## 4.1. Experimental Setup

**MATS-GPT2.** In the smaller version, we employ GPT2 with 125M parameters as the LLM. We use CLAP's audio encoder and language encoder. The audio is sampled at 44.1 kHz and converted into a log Mel spectrograms with 64 Mel bins, a hop size of 320 ms, and a window size of 1024 ms. All audio files are randomly truncated to 7 seconds in length with CLAP audio encoder. The mapping module uses a 8-layer transformer with a prefix length of 40. Training is conducted using the AdamW optimizer with a learning rate of $5 \times 10^{-5}$, a linear learning rate scheduler with 2000 warmup steps. The batch size is set to 128. For the LoRA configuration, we set the rank to 8, the scaling factor to 4, and the dropout rate to 0.1. MATS-GPT2 underwent about 25 hours of training over 90,000 iterations on 2 A100 GPUs. The hyperparameters are configured as follows: $\sigma = 0.015$, $K = 100$, $L = 32$, $\tau = 0.1$ and $\lambda = 0.3$ (Equation 7).

**MATS-LLaMA.** In the larger version, we employ the LLaMA-7B model as the LLM. We use AdamW optimizer with a learning rate of $3 \times 10^{-5}$, a cosine annealing learning rate scheduler with 3000 warmup steps. The batch size is set to 96. MATS-LLaMA underwent about 35 hours of training over 360,000 iterations. All other settings remain consistent with those of MATS-GPT2.

**MATS-Audio.** We explore an audio version of MATS, supervised by audio. MATS-Audio consists of the CLAP audio encoder, the mapper module and LLaMA-7B. It employs the same training configurations as MATS-LLaMA,

with a different training set as outlined in Appendix B.2.

## 4.2. Main Results

We evaluate the close-ended and open-ended capabilities of MATS using 16 downstream tasks. Table S10 in Appendix summarizes all test benchmarks and corresponding evaluation metrics. Notably, for classification task, we adopt the text-matching setup following (Gong et al., 2024), where the CLIP language encoder is used to encode both the answer set and model's predictions. The answer with the highest similarity to the prediction is selected as the final result.

### 4.2.1. CLOSE-ENDED RESULTS

**Zero-shot Audio Classification.** Since MATS performs audio classification solely based on GPT-generated descriptions, its performance is evaluated under zero-shot setting. Table 1 summarizes the comparative results across both sound and music types. As shown in Table 1, MATS achieves SOTA performance in the zero-shot setting on FSD50K, DCASE, TUT, VGG and BJO benchmarks, with improvements of $1\%, 10\%, 14\%, 6\%$ and $3\%$. Remarkably, MATS performs comparably with SALMONN (Tang et al., 2024) and Qwen-Audio Chat (Bai et al., 2023), both of which are trained on a significant amount of audio-language pairs. These results validate the efficacy of our approach for audio classification, highlighting the excellent audio perception capabilities of MATS in a zero-shot scenario.

**Audio Captioning.** Table 2 summarizes the comparative results for audio captioning tasks on AudioCaps, Clotho, and MusicCaps benchmarks. We can observe that: (1) Compared to text-only supervised audio captioning models, which are specifically tailored for audio captioning and restricted to the *Sound* type, our model demonstrates enhanced versatility by effectively addressing a broader spectrum of audio-relevant tasks across both *Sound* and *Music* type. Despite its broader applicability, our model achieves superior CIDEr and SPIDEr scores on AudioCaps and a higher SPICE score on Clotho. The performance gains can be largely attributed to the proposed **Santa** mechanism, which effectively mitigates the audio-language modality gap within CLAP. (2) Compared to audio-supervised models, MATS-LLaMA exhibits comparable performance despite being training only on text data. Also, MATS-GPT2 surpasses the music captioning model LP-MusicCaps (Doh et al., 2023) by **9.9%** on the ROUGH-L metric. These results underscore the effective audio captioning capabilities of MATS under text-only supervision.

**Simple AQA.** The ClothoAQA benchmark primarily consists of responses limited to `yes` or `no` answers (evaluated using the B-ACC metric), framing the task largely as a classification problem. As shown in Table 1, MATS-LLaMA outperforms the recent model Qwen-Audio (Chu et al., 2023)

Table 1. Comparision results on audio classification benchmarks and ClothoAQA benchmark. $^{ZS-}$ and $^{\dagger}$ indicate weakly zero-shot and supervised settings, respectively. Results without any annotation represent zero-shot setting. **Note:** the results of SALMONN and Qwen-Audio-Chat are from GAMA (Ghosh et al., 2024).

| Audio Type | Sound | | | | | | | | Music | |
|---|---|---|---|---|---|---|---|---|---|---|
| Model | ESC-50 (ACC) | FSD50K (mAP) | DCASE (ACC) | TUT (ACC) | VGG (ACC) | US8K (ACC) | ClothoAQA (ACC\|B-ACC) | | GTZAN (ACC) | BJO (ACC) |
| CLAP (Elizalde et al., 2023) | 0.83 | 0.30 | 0.30 | 0.30 | - | 0.73 | - | - | 0.25 | 0.30 |
| Pengi (Deshmukh et al., 2023) | **0.92** | 0.47 | 0.33 | 0.35 | - | 0.72 | - | $0.65^{\dagger}$ | 0.35 | 0.62 |
| LTU (Gong et al., 2024) | $0.83^{ZS-}$ | $0.46^{\dagger}$ | $\mathbf{0.46}^{ZS-}$ | 0.33 | $\underline{0.50}^{\dagger}$ | - | - | - | - | 0.70 |
| LTU-AS (Gong et al., 2024) | $0.81^{ZS-}$ | - | - | - | - | - | - | - | 0.50 | - |
| GAMA (Ghosh et al., 2024) | $0.83^{ZS-}$ | $\underline{0.48}^{\dagger}$ | $0.38^{ZS-}$ | 0.22 | $\mathbf{0.52}^{\dagger}$ | - | - | $0.72^{\dagger}$ | 0.14 | 0.70 |
| Qwen-Audio (Chu et al., 2023) | - | - | - | $\mathbf{0.65}^{\dagger}$ | - | - | $0.58^{\dagger}$ | $0.75^{\dagger}$ | - | - |
| Audio Flamingo (Kong et al., 2024) | - | $\mathbf{0.70}^{\dagger}$ | - | - | - | **0.75** | - | $\mathbf{0.87}^{\dagger}$ | **0.68** | - |
| SALMONN (Tang et al., 2024) | 0.16 | 0.22 | 0.18 | 0.08 | 0.23 | - | - | - | 0.10 | 0.25 |
| Qwen-Audio Chat (Bai et al., 2023) | 0.72 | 0.40 | 0.32 | 0.17 | 0.18 | - | - | - | 0.42 | 0.51 |
| **MATS-GPT2 (Ours)** | 0.86 | 0.48 | 0.43 | 0.49 | 0.29 | 0.67 | 0.56 | 0.66 | 0.58 | 0.70 |
| **MATS-LLaMA (Ours)** | 0.88 | 0.45 | 0.43 | 0.47 | 0.29 | 0.67 | 0.66 | 0.76 | 0.63 | **0.73** |
| MATS-Audio | 0.70 | $0.43^{\dagger}$ | $0.55^{\dagger}$ | $0.61^{\dagger}$ | $0.42^{\dagger}$ | 0.39 | $0.67^{\dagger}$ | $0.76^{\dagger}$ | 0.29 | 0.46 |

Table 2. Comparison results for audio captioning tasks on AudioCaps, Clotho and MusicCaps benchmarks.

| | Method | AudioCaps | | | Clotho | | | MusicCaps | |
|---|---|---|---|---|---|---|---|---|---|
| | | CIDEr | SPICE | SPIDEr | CIDEr | SPICE | SPIDEr | ROUGH-L | BLUE4 |
| Audio Supervision (AS) | Pengi (Deshmukh et al., 2023) | **0.752** | 0.182 | 0.467 | 0.416 | 0.126 | 0.271 | - | - |
| | LTU (Gong et al., 2024) | - | 0.170 | - | - | 0.119 | - | - | - |
| | GAMA (Ghosh et al., 2024) | - | 0.185 | - | - | **0.135** | - | - | - |
| | LTU-AS (Gong et al., 2024) | - | 0.150 | - | - | - | - | - | - |
| | SALMONN (Tang et al., 2024) | - | - | 0.485 | - | - | - | 21.5 | 6.1 |
| | Audio Flamingo (Kong et al., 2024) | - | - | **0.502** | 0.465 | - | - | - | - |
| | LP-MusicCaps (Doh et al., 2023) | - | - | - | - | - | - | 13.0 | 0.7 |
| Text Supervision (TS) | NoAudioCaptioning (Deshmukh et al., 2024) | 0.697 | 0.178 | 0.437 | 0.379 | 0.132 | 0.256 | - | - |
| | PromptAAC (Zhang et al., 2024) | 0.644 | 0.156 | 0.400 | 0.403 | 0.119 | 0.261 | - | - |
| | DRCap (Li et al., 2024) | 0.718 | **0.186** | 0.452 | 0.438 | 0.133 | **0.285** | - | - |
| | **MATS-GPT2 (Ours)** | 0.676 | 0.164 | 0.420 | 0.413 | 0.124 | 0.269 | **22.9** | 4.5 |
| | **MATS-LLaMA (Ours)** | 0.735 | 0.171 | 0.453 | 0.431 | 0.134 | 0.282 | 18.7 | 3.2 |
| AS | MATS-Audio | 0.704 | 0.171 | 0.438 | 0.448 | 0.137 | 0.293 | 21.7 | 5.4 |

on ClothoAQA, achieving $8\%$ higher ACC score. This result highlights the enhanced audio comprehensive capabilities of MATS-LLaMA after scaling up. However, MATS shows slightly lower performance on the B-ACC metric compared to Audio Flamingo (Kong et al., 2024). Notably, Audio Flamingo leverages audio supervision during training and employs a sliding window to extract audio features. While this enhances its performance, it also introduces significant computational overhead. In contrast, MATS-LLaMA operates under text-only supervision, offering a more computationally efficient alternative. This trade-off between efficiency and performance underscores the practical considerations in model design for real-world applications, where computational cost is often a critical factor.

### 4.2.2. OPEN-ENDED RESULTS

To further evaluate our model's understanding and reasoning capabilities in complex audio QA tasks, we conducted experiments on AIR-Bench Chat (Yang et al., 2024) and MMAU (Sakshi et al., 2024) benchmarks. The results are presented in Table 4 and Table 3. **(1)** On AIR-Bench, under text-only supervision, MATS-LLaMA achieves performance comparable to Qwen-Audio (Chu et al., 2023), ranking third-highest on *Sound* type and fourth-highest on *Music* type. We argue that CLAP struggles to extract fine-grained audio information, such as the exact number of occurrences. This limitation makes MATS-LLaMA more prone to errors when responding to fine-grained questions on AIR-Bench. **(2)** For the MMAU, the QA format requires LALMs to select from a set of provided options. We use the following prompt: {question} + Select one option: {options} Respond with exactly one of the options above. Compared to AIR-Bench, this option-based format offers additional contextual cues, partially mitigating CLAP's limitations in extracting detailed information. Consequently, our model outperforms Qwen-Audio-Chat (Bai et al., 2023) by about $3\%$ in the sound type and ranks second only to the Qwen2-Audio-Instruction (Chu et al., 2024) in other audio types. Notably, our model even surpasses larger-scale models like Qwen-Audio-Chat and SALMONN.

*Table 3.* The results of our model on the MMAU datasets across the sound and music categories.

| | Method | Size | Sound | | Music | | Avg | |
|---|---|---|---|---|---|---|---|---|
| | | | Test-mini | Test | Test-mini | Test | Test-mini | Test |
| AS | Qwen2-Audio-Instruction (Chu et al., 2024) | 8.4B | 55.0 | 45.9 | **51.0** | **53.3** | **49.2** | **52.5** |
| | Qwen-Audio-Chat (Bai et al., 2023) | 8.4B | **55.3** | 56.7 | 44.0 | 40.9 | 43.1 | 41.9 |
| | SALMONN (Tang et al., 2024) | 13B | 41.0 | 40.3 | 34.8 | 33.8 | 33.7 | 32.8 |
| | GAMA (Ghosh et al., 2024) | 7B | 41.4 | 45.4 | 32.3 | 30.8 | 30.9 | 31.8 |
| | MuLLaMa (Liu et al., 2024b) | 7B | 40.8 | 44.8 | 32.6 | 30.6 | 31.9 | 30.7 |
| | GAMA-IT (Ghosh et al., 2024) | 7B | 43.2 | 43.2 | 28.4 | 28.0 | 30.2 | 29.0 |
| | LTU-AS (Gong et al., 2024) | 7B | 23.4 | 25.0 | 9.1 | 10.5 | 17.7 | 18.9 |
| | LTU (Gong et al., 2024) | 7B | 22.5 | 25.9 | 9.7 | 12.8 | 16.9 | 18.5 |
| | Pengi (Deshmukh et al., 2023) | 323M | 6.1 | 8.0 | 2.9 | 3.1 | 3.4 | 4.2 |
| TS | **MATS-GPT2 (Ours)** | 370M | 7.8 | 7.2 | 14.1 | 11.7 | 10.5 | 9.8 |
| | **MATS-LLaMA (Ours)** | 7B | 52.3 | **59.8** | 44.9 | 42.9 | 43.3 | 44.5 |
| AS | MATS-Audio | 7B | 51.7 | 57.4 | 43.7 | 41.5 | 42.2 | 43.8 |

Furthermore, compared to MATS-Audio, MATS-LLaMA achieves comparable results on open-ended tasks, further validating the effectiveness of the text-supervised strategy for training LALMs.

*Table 4.* The results of our model on the AIR-Bench Chat Benchmark across the sound and music categories.

| | Method | Sound | Music |
|---|---|---|---|
| AS | Qwen2-Audio (Chu et al., 2024) | **6.99** | **6.79** |
| | Qwen-Audio-Turbo (Bai et al., 2023) | 6.59 | 5.98 |
| | SALMONN (Tang et al., 2024) | 6.28 | 5.95 |
| | Qwen-Audio (Chu et al., 2023) | 6.95 | 5.52 |
| | Gemini-1.5-pro (Team et al., 2024) | 5.49 | 5.06 |
| | BLSP (Wang et al., 2023) | 5.55 | 5.08 |
| | Pandagpt (Su et al., 2023) | 5.46 | 5.06 |
| | Next-gpt (Wu et al., 2024) | 4.76 | 4.18 |
| | SpeechGPT (Zhang et al., 2023) | 0.95 | 0.95 |
| | Macaw-LLM (Lyu et al., 2023) | 1.01 | 0.91 |
| TS | **MATS-GPT2 (Ours)** | 4.33 | 3.31 |
| | **MATS-LLaMA (Ours)** | 6.43 | 5.76 |
| AS | MATS-Audio | 6.25 | 5.77 |

### 4.3. Ablation

In this subsection, we conduct ablation studies to validate the effectiveness of our method. Further analysis of the hyperparameters in Equation (7), including $K$, $L$, $M$, $\tau$, $\lambda$, and the number of mapper layers, is provided in Appendix C.2

**MATS-GPT2 vs MATS-LLaMA.** As shown in Table 1 and Table 2 , MATS-GPT2 achieves performance comparable to MATS-LLaMA on close-ended tasks. However, as shown in Table 4 and Table 3, MATS-GPT2 exhibits significantly weak performance on open-ended tasks. These results indicate that a smaller LLM like GPT2 is sufficient for handling basic audio perception tasks, but it struggles with complex open-ended tasks that require advanced audio understanding and reasoning. In contrast, larger LLMs are better equipped to capture the depth and complexity necessary for audio reasoning and comprehension, making them indispensable

for open-ended tasks.

**MATS-LLaMA vs MATS-Audio.** As shown in Table 1, MATS-LLaMA achieves performance comparable to MATS-Audio in the supervised setting. However, under the zero-shot setting, MATS-LLaMA significantly outperforms MATS-Audio, highlighting the heavy reliance of audio-supervised models on audio-language pairs for achieving superior performance. Moreover, as shown in Table 4 and Table 3, MATS-LLaMA achieves performance on par with MATS-Audio in open-ended scenarios, further highlighting the cost-effectiveness and efficiency of our approach.

**Santa.** Table 6 presents the ablation study of our proposed **Santa** mechanism. The study evaluates the impact of excluding various components, including k-means (w/o K-means), memory bank (w/o MB), audio embedding (w/o AE) and Gaussian noise (w/o GN). As shown in Table 6, removing any component of Santa results in a performance drop, highlighting their collective effectiveness. Notably, the absence of Gaussian noise (w/o GN) during training and the exclusion of audio embeddings (w/o AE) during inference lead to significant performance decline. These results emphasize the critical role of incorporating Gaussian noise during training and retaining audio embeddings during inference to achieve robust performance.

**Santa vs other modal-transfer mechanism.** To demonstrate the superior effectiveness of the Santa mechanism, we compare Santa with other modal-transfer methods, including the memory-based mechanism in DRCap (Li et al., 2024), the noise-based mechanism in NoAudioCaptioning (Deshmukh et al., 2024) and PromptAAC (Zhang et al., 2024). As shown in Table 5, Santa consistently outperforms the competing methods, validating its strength in bridging the audio-language modality gap. In DRCap, the original audio embeddings are completely discarded during inference, making its performance highly dependent on the alignment between the memory bank and the test benchmark. However, in a multi-task setting, the memory bank must accommo-

*Table 5.* Performance Comparison between Santa and Other Modality Transfer Methods on Close-Ended and Open-Ended Benchmarks

| Benchmark | ESC-50 | AudioCaps | AIRBench-Sound | AIRBench-Music |
|---|---|---|---|---|
| Metric | ACC | CIDEr | GPT4 | GPT4 |
| MATS-NoAudioCaptioning | 0.88 | 0.698 | 6.15 | 5.71 |
| MATS-PromptAAC | 0.77 | 0.593 | 6.07 | 5.28 |
| MATS-DRCap | 0.84 | 0.619 | 5.83 | 5.29 |
| **MATS-LLaMA** | **0.88** | **0.735** | **6.43** | **5.76** |

date a diverse range of benchmarks rather than a single one, which introduces noise into the retrieval process and projects the audio embeddings into less relevant textual spaces, leading to performance drop. NoAudioCaptioning, on the other hand, employs a pure noise-injection strategy. While simple, this approach fails to effectively reduce the modality gap within CLAP, as analyzed in detail in Appendix C.1, resulting in limited performance gains. PromptAAC adopts an augmentation-based approach that involves injecting noise and substituting similar language inputs. It retrieves audio events by matching audio embeddings with language embeddings derived from 527 predefined audio labels in AudioSet. However, the limited variety of audio events restricts the diversity of the retrieved information, resulting in inferior performance compared to Santa.

**The impact of Gaussian noise variance** $\sigma$. Table 7 investigates the impact of the variance $\sigma$ of injected noise during training. MATS achieves the best performance at $\sigma = 0.015$. We argue that a small variance fails to effectively mitigate the audio-language modality gap within CLAP, while an excessively large variance impairs the model's ability to capture useful input information.

*Table 6.* Ablation Study of Santa on AudioCaps benchmark.

| Method | CIDEr | SPICE | SPIDEr |
|---|---|---|---|
| MATS-LLaMA | **0.735** | **0.171** | **0.453** |
| - w/o K-means | 0.703 | 0.171 | 0.437 |
| - w/o MB | 0.698 | 0.170 | 0.434 |
| - w/o AE | 0.343 | 0.114 | 0.229 |
| - w/o GN | 0.384 | 0.121 | 0.253 |

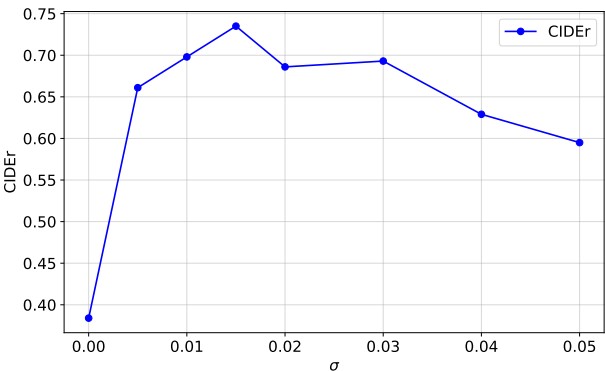

*Table 7.* The impact of $\sigma$ on AudioCaps benchmark.

## 5. Conclusion

In this paper, we propose MATS, an audio-language multimodal LLM tuned solely on text data. Leveraging an efficient text-only training strategy, MATS establishes the semantic connection between the LLM and audio-language contrastive mode, substantially reducing data and training costs while improving zero-shot performance. To address the modality gap issue, we further introduce a modal-transfer strategy, Santa, which effectively balances audio embeddings and augmented language embeddings. Comprehensive experiments show that MATS achieves performance comparable to, and in some cases surpassing, LALMs trained on large-scale audio-language paired datasets.

## Acknowledgements

This work is partially supported by National Natural Science Foundation of China (NSFC): 62376259 and 62306301, National Postdoctoral Program for Innovative Talents under Grant BX20220310.

## Impact Statement

This paper presents work whose goal is to advance the field of Large Audio-Language Models. There are many potential societal consequences of our work, none which we feel must be specifically highlighted here.

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

## A. Proof of Theorem 3.1

In this section, we provide a proof for Theorem 3.1.

*Proof.* Let $R_{\mathrm{tr}}(h) = \mathbb{E}_{(z^t,y)\sim p_{\mathcal{T}}(z^t,y)} |h(z^t) - y|$ denote the generalization error on the train set. According to triangle inequality, we can drive that:

$$
\begin{aligned}
R_{\mathrm{te}}(h) &= R_{\mathrm{te}}(h) - R_{\mathrm{tr}}(h) + R_{\mathrm{tr}}(h) - \hat{R}_{\mathrm{tr}}(h) + \hat{R}_{\mathrm{tr}}(h) \\
&\leq \left|R_{\mathrm{te}}(h) - R_{\mathrm{tr}}(h)\right| + \left|R_{\mathrm{tr}}(h) - \hat{R}_{\mathrm{tr}}(h)\right| + \hat{R}_{\mathrm{tr}}(h)
\end{aligned}
\tag{8}
$$

$$
\begin{aligned}
\left|R_{\mathrm{te}}(h) - R_{\mathrm{tr}}(h)\right| &= \left|\mathbb{E}_{(z^a,y)\sim p_{\mathcal{A}}(z^a,y)}|h(z^a) - y| - \mathbb{E}_{(z^t,y)\sim p_{\mathcal{T}}(z^t,y)}|h(z^t) - y|\right| \\
&= \left|\sum_v^V p_{\mathcal{A}}(y) \cdot \mathbb{E}_{(z^a)\sim p_{\mathcal{A}}(z^a|y)}|h(z^a) - y| - \sum_v^V p_{\mathcal{T}}(y) \cdot \mathbb{E}_{(z^t)\sim p_{\mathcal{T}}(z^t|y)}|h(z^t) - y|\right|
\end{aligned}
\tag{9}
$$

Assuming that the class distributions for training and testing are consistent, we can obtain $p_{\mathcal{A}}(y) = p_{\mathcal{T}}(y)$, then we can achieve that:

$$
\begin{aligned}
\left|R_{\mathrm{te}}(h) - R_{\mathrm{tr}}(h)\right| &= \left|\sum_v^V p_{\mathcal{A}}(y) \cdot (\mathbb{E}_{(z^a)\sim p_{\mathcal{A}}(z^a|y)}|h(z^a) - y| - \mathbb{E}_{(z^t)\sim p_{\mathcal{T}}(z^t|y)}|h(z^t) - y|)\right| \\
&\leq \max_V \left|\mathbb{E}_{(z^a)\sim p_{\mathcal{A}}(z^a|y)}|h(z^a) - y| - \mathbb{E}_{(z^t)\sim p_{\mathcal{T}}(z^t|y)}|h(z^t) - y|\right| \\
&\leq \max_V \mathrm{disc}_{L_1}(p_{\mathcal{A}}(z^a|y), p_{\mathcal{T}}(z^t|y))
\end{aligned}
\tag{10}
$$

$\left|R_{\mathrm{tr}}(h) - \hat{R}_{\mathrm{tr}}(h)\right|$ follows by a standard application Vapnik-Chervonenkis (Vapnik, 1998) theory to bound the true $R_{\mathrm{tr}}(h)$ by its empirical estimate $\hat{R}_{\mathrm{tr}(h)}$. Namely, if the $D_{\mathrm{tr}}$ is a $N$-size .i.i.d. samples, then with probability exceeding $1 - \delta$,

$$
\left|R_{\mathrm{tr}}(h) - \hat{R}_t r(h)\right| \leq \sqrt{\frac{8}{N}\left(2d \log \sqrt{2NV} + \log \frac{2}{\delta}\right)}
\tag{11}
$$

At the last, we can achieve that:

$$
R_{te}(h) \leq \hat{R}_{tr}(h) + \max_v \left(\mathrm{disc}_{L_1}\left(p_{\mathcal{A}}(z^a|y), p_{\mathcal{T}}(z^t|y)\right)\right) + \sqrt{\frac{8}{N}\left(2d \log \sqrt{2NV} + \log \frac{2}{\delta}\right)}.
\tag{12}
$$

$\square$

## B. Dataset

### B.1. The train dataset of MATS-GPT2 and MATS-LLaMA

Table S8 provides a detailed overview of the training dataset, `AudioTIA`. In this dataset, the inputs are primarily composed of descriptions of audio contents and textual instrucrions. `AudioTIA` is divided into close-ended and open-ended tasks, focusing on two audio types: Sound and Music. Specifically, the close-ended dataset, consisting of about 1.5M samples, primarily includes classification tasks (58.7%), captioning tasks (40%), and the remaining portion comprises simple audio question-answering tasks. In contrast, the open-ended dataset, comprising about 3.8M samples, focuses on complex audio question-answering tasks. The close-ended dataset is designed to establish the model's foundational audio perception capabilities, and the open-ended dataset focuses on enhancing the model's capacity for advanced audio understanding and reasoning.

*Table S8.* The statics of the `AudioTIA` dataset.

| Audio Type | Task | Datasets | #Samples | Percentage |
|---|---|---|---|---|
| **Close-Ended Audio Text Instruction Answering Data (∼1.5M)** | | | | |
| Sound | CLS | AudioSet (Gemmeke et al., 2017), FSD50k (Fonseca et al., 2021), VGGSound (VGG) (Chen et al., 2020), DCASE (Mesaros et al., 2017), ECS-50 (Piczak, 2015), UrbanSound8K(US8K) (Salamon et al., 2014) TUT Acoustic Scenes 2017(TUT) (Mesaros et al., 2016), Sound Events for Surveillance Applications(SESA) (Spadini, 2019) VS (Gong et al., 2022), CREMA-D (Cao et al., 2014) RAVDESS (Livingstone & Russo, 2018) | ∼385K | ∼25.6% |
| | CAP | WavCaps (Mei et al., 2024), Macs (Martín-Morató & Mesaros, 2021), AudioCaps (Kim et al., 2019), Clotho-v2 (Drossos et al., 2020) | ∼518K | ∼34.5% |
| | AQA | Clotho-AQA (Lipping et al., 2022) | ∼20K | ∼1.3% |
| Music | CLS | GTZAN (Sturm, 2014), Beijing Opera (BJO) (Tian et al., 2014), OpenAQA (Ghosh et al., 2024) | ∼497K | ∼ 33.1% |
| | CAP | OpenAQA(Ghosh et al., 2024), MusicCaps (Agostinelli et al., 2023), Song Describer dataset (SDD) (Manco et al., 2023) | ∼30K | ∼5.5% |
| **Open-Ended Audio Text Instruction Answering Data (∼3.8M)** | | | | |
| Sound | AQA | OpenAQA (Gong et al., 2024) | ∼2547K | ∼66.8% |
| Music | AQA | MusicQA (Liu et al., 2024a) | ∼1207K | ∼33.2% |

## B.2. The train dataset of MATS-Audio

Table S9 presents the training dataset of MATS-Audio, referred to as `AudioAIA`. The data format for `AudioAIA` is (`audio`, `instruction`, `answer`), which differs from the input format of `AudioTIA` with (`text`, `instruction`, `answer`). The `AudioAIA` dataset is derived from `AudioTIA`, excluding any subsets used exclusively as test data. Overall, the `AudioAIA` dataset comprises approximately 1.5M samples for close-ended tasks and 2.6M samples for open-ended tasks.

*Table S9.* The statics of the `AudioAIA` dataset.

| Audio Type | Task | Datasets | #Samples | Percentage |
|---|---|---|---|---|
| **Close-Ended Audio Instruction Answering Data (∼1.5M)** | | | | |
| Sound | CLS | AudioSet (Gemmeke et al., 2017), FSD50k (Fonseca et al., 2021), VGGSound (VGG) (Chen et al., 2020), DCASE (Mesaros et al., 2017), TUT Acoustic Scenes 2017(TUT) (Mesaros et al., 2016), Sound Events for Surveillance Applications(SESA) (Spadini, 2019) | ∼331K | ∼21.9% |
| | CAP | WavCaps (Mei et al., 2024), Macs (Martín-Morató & Mesaros, 2021), AudioCaps (Kim et al., 2019), Clotho-v2 (Drossos et al., 2020) | ∼517K | ∼34.2% |
| | AQA | Clotho-AQA (Lipping et al., 2022) | ∼6.1K | ∼0.4% |
| Music | CLS | OpenAQA (Ghosh et al., 2024) | ∼630K | ∼41.7% |
| | CAP | OpenAQA(Ghosh et al., 2024), MusicCaps (Agostinelli et al., 2023), | ∼28.3K | ∼1.9% |
| **Open-Ended Audio Instruction Answering Data (∼2.6M)** | | | | |
| Sound | AQA | OpenAQA (Gong et al., 2024) | ∼2467K | ∼95.4% |
| Music | AQA | MusicQA (Liu et al., 2024a) | ∼118K | ∼4.6% |

## B.3. The test benchmark of MATS

Table S10 provides a detailed overview of the test benchmark and corresponding evaluation metrics. The benchmark covers a range of audio event classification tasks, such as ESC-50 (Piczak, 2015) (single-label classification), FSD50K (Fonseca et al., 2021) (multi-label classification), DCASE (Mesaros et al., 2017), TUT (Mesaros et al., 2016), VGG (Chen et al., 2020), US8K (Salamon et al., 2014), GTZAN (Sturm, 2014) and BJO (Tian et al., 2014), as well as audio captionning

tasks like AudioCaps (Kim et al., 2019) and Clotho-v2 (Drossos et al., 2020). It also encompasses music captioning tasks including MusicCaps (Agostinelli et al., 2023), simple QA tasks on ClothoAQA, and complex QA tasks such as AIR-Bench Chat (Yang et al., 2024) and MMAU (Sakshi et al., 2024) benchmarks.

*Table S10.* The test benchmarks and evaluation metrics are presented. B-ACC represents binary accuracy. **Note:** All evaluation metrics are designed such that higher values indicate better performance.

| Close-Ended Audio Instruction Answering Test Data | | | |
|---|---|---|---|
| Audio Type | Task | Benchmark | Metric ↑ |
| Sound | CLS | ECS-50 (Piczak, 2015) | ACC |
| | CLS | FSD50K (Fonseca et al., 2021) | mAP |
| | CLS | DCASE (Mesaros et al., 2017) | ACC |
| | CLS | TUT (Mesaros et al., 2016) | ACC |
| | CLS | VGG (Chen et al., 2020) | ACC |
| | CLS | US8K (Salamon et al., 2014) | ACC |
| | CAP | AudioCaps (Kim et al., 2019) | CIDEr\|SPICE\|SPIDEr |
| | CAP | Clotho-v2 (Drossos et al., 2020) | CIDEr\|SPICE\|SPIDEr |
| | AQA | Clotho-AQA (Lipping et al., 2022) | ACC\|B-ACC |
| Music | CLS | GTZAN (Sturm, 2014) | ACC |
| | CLS | BJO (Tian et al., 2014) | ACC |
| | CAP | MusicCaps (Agostinelli et al., 2023) | BLUE4\|ROUGH-L |
| **Open-Ended Audio Instruction Answering Test Data** | | | |
| Sound | AQA | AIR-Bench Chat (Yang et al., 2024) | GPT4 |
| | AQA | MMAU (Sakshi et al., 2024) | ACC |
| Music | AQA | AIR-Bench Chat (Yang et al., 2024) | GPT4 |
| | AQA | MMAU (Sakshi et al., 2024) | ACC |

### B.4. The design for generating audio descriptions in close-ended tasks.

In this part, we describe the pipeline and prompt used with ChatGPT (Achiam et al., 2023) to generate audio descriptions for various classification task datasets, including ESC-50 (Piczak, 2015), FSD50K (Fonseca et al., 2021), VGG (Chen et al., 2020), ECS-50 (Piczak, 2015), US8K (Salamon et al., 2014), TUT (Mesaros et al., 2016), SESA (Spadini, 2019), VS (Gong et al., 2022), CREMA-D (Cao et al., 2014), and RAVDESS (Livingstone & Russo, 2018) for sound classification, and GTZAN (Sturm, 2014) and BJO (Tian et al., 2014) for music classification. Figure S4 provides an overview of the pipeline. The pipeline involves supplying GPT-4o-mini with a set of example audio descriptions corresponding to a few labels, followed by a prompt to guide the generation process. GPT-4o-mini then produces descriptions for all provided labels. The final set of generated descriptions, paired with their corresponding labels, serves as input for MATS.

### B.5. The instruction templates for training MATS

Recent studies (Deshmukh et al., 2023; Tang et al., 2024) have demonstrated that instruction fine-tuning enhances the instruction-following abilities and comprehension capabilities of audio-language multimodal models. Building on this insight, we adopt instruction fine-tuning when training MATS. Table S11 presents the instruction templates designed for all tasks in our study.

## C. More Experimental Results

### C.1. Visualization Analysis on Audio-Language Modality Gap

Figure 3 presents a visualization analysis of the audio-language modality gap. As shown in Figure 3 $a$), a noticeable gap (Liang et al., 2022) remains between audio embeddings and language embeddings of CLAP. In Figure 3 $b$), introducing Gaussian noise reduces the embedding gap to some extent; however, the language and audio embeddings remain distributed near opposite ends. Similarly, in Figure 3 $c$), the memory-based method shows a comparable effect to $b$). As previously noted, this method completely replaces audio embeddings with language embeddings, resulting in the processed audio

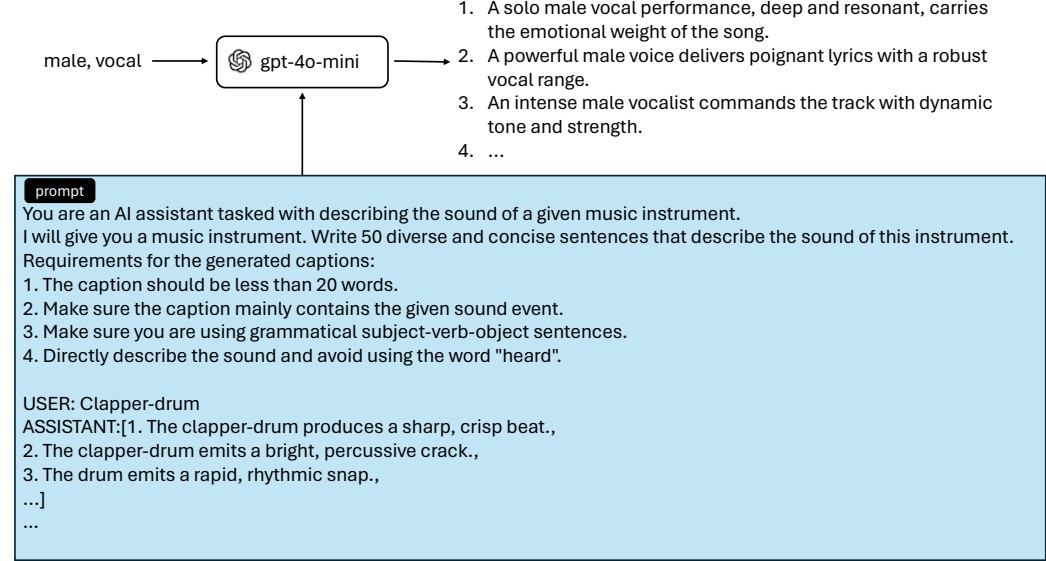

*Figure S4.* Pipeline And Prompt of generating audio descriptions.

*Table S11.* The instruction template of every task. CLS: Audio Event Classification; CAP: Audio Captioning; AQA: Audio Question Answer.

| Task | Input text | Input instruction format | Output answer format |
|------|-----------|--------------------------|----------------------|
| CLS | GPT generated description | "Classify events in the audio clip." | label a, ... |
| Sound CAP | caption | "Generate audio caption." | caption |
| Music CAP | caption | "Generate music caption." | caption |
| AQA | generated caption | Question | answer |

features losing audio-specific information and becoming overly concentrated, thereby reducing the model's generalization ability. In contrast, Figure 3 *d*) illustrates that our **Santa** further integrates the two modalities into a more unified and cohesive distribution.

## C.2. Hyperparameter Analysis

**The impact of the different temperature $\tau$.** In Table S12, we analyze the impact of different $\tau$ values on the model's performance. Our model achieves the best performance when $\tau = 0.1$. When $\tau$ is too small, our algorithm tends to select only the most similar embedding, which limits the representational power of language embeddings and leads to a decrease in model performance. Conversely, when $\tau$ is too large, the model may capture irrelevant augmented language embeddings with low relevance.

*Table S12.* The performance of different $\tau$ on AudioCaps.

| $\tau$ | 1/150 | 1/80 | 1/40 | 1/20 | 1/10 | 1 |
|--------|-------|------|------|------|------|---|
| CIDEr | 0.731 | 0.734 | 0.732 | 0.734 | **0.735** | 0.731 |

*Table S13.* The performance of different $L$ on AudioCaps.

| $L$ | 4 | 8 | 16 | 32 | 64 |
|-----|---|---|----|----|----|
| CIDEr | 0.728 | 0.730 | 0.729 | **0.735** | 0.734 |

**The impact of different top $L$.** Table S13 presents the performance of our method with different values of $L$. When $L$ is too large, the model may capture irrelevant augmented language embeddings with low relevance. Conversely, as $L$ approaches 1, the model loses the benefits of multiple language representations. We set $L = 32$ to strike a balance.

**The impact of the different balancing parameter** $\lambda$**.** In this part, we conducted experiments with different values of $\lambda$, as shown in Figure S5. The results clearly demonstrate that the best performance is achieved when both language and audio are integrated simultaneously, and the performance is relatively insensitive to the variations in $\lambda$. However, when audio is entirely removed, the model's performance drops significantly, further validating the effectiveness of the balancing strategy within the Santa mechanism.

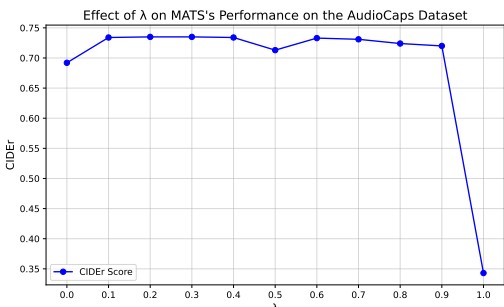

*Figure S5.* The effect of different $\lambda$ on AudioCaps.

**Hyper-Variations in LoRA.** Table S14 and Table S15 present the performance variations of MATS on the AudioCaps test benchmark under different LoRA ranks and scaling factors. When the rank or scaling factor is too small, the model undergoes insufficient updates, leading to underfitting. Conversely, excessively large ranks or scaling factors result in overly aggressive updates, increasing model complexity and causing overfitting.

*Table S14.* The performance of MATS with different ranks of LoRA on AudioCaps.

| Rank | 4 | 8 | 16 |
|---|---|---|---|
| CIDEr | 0.706 | **0.735** | 0.702 |

*Table S15.* The performance of MATS with different scaling factors of LoRA on AudioCaps.

| $\frac{\alpha}{Rank}$ | 2 | 4 | 8 |
|---|---|---|---|
| CIDEr | 0.674 | **0.735** | 0.682 |

**The impact of the different number of** $\mathcal{M}$**.** Table S16 shows that the performance of our model initially increases and then decreases as the size of the memory bank $\mathcal{M}$ grows. The initial improvement is attributed to the increasing richness of the provided language embedding information. However, as the memory bank continues to expand, the fixed value of $L$ limits the model's ability to effectively utilize diverse language embeddings.

*Table S16.* The performance of MATS with the different number of memory bank $\mathcal{M}$ on AudioCaps.

| $M$ | 1000 | 2000 | 3000 | 3500 | 4000 | 4500 | 5000 | 5172 | 5500 | 6000 | 7000 |
|---|---|---|---|---|---|---|---|---|---|---|---|
| CIDEr | 0.710 | 0.700 | 0.714 | 0.716 | 0.719 | 0.705 | 0.708 | **0.735** | 0.705 | 0.700 | 0.700 |

**The impact of k-means cluster size** $K$**.** Table S17 investigates the impact of the k-means cluster size $K$ in the Santa mechanism. As shown, regardless of the cluster size, the model consistently outperforms noise-based and memory-based methods, indicating that our Santa mechanism is robust to the choice of the k-means cluster number. We set $K = 100$ in this work.

**The number of Mapper'layers.** We further evaluated the impact of varying the number of Transformer layers on the model's performance on AudioCaps test set. As shown in Table S18, the model achieves optimal performance with 8 layers. When the number of layers is fewer than 8, the model's capacity is insufficient for effective learning, leading to underfitting. Conversely, increasing the number of layers beyond 8 results in overfitting, thereby reducing generalization ability. Based on these findings, we adopt an 8-layer Transformer in this work.

*Table S17.* Ablation study of MATS-LLaMA on the cluster number $K$ in the Santa mechanism, evaluated on AudioCaps benchmark.

| $K$ | CIDEr | SPICE | SPIDEr |
|-----|-------|-------|--------|
| 10 | 0.731 | 0.170 | 0.450 |
| 30 | 0.733 | 0.170 | 0.452 |
| 60 | 0.734 | 0.171 | 0.452 |
| 100 | **0.735** | **0.171** | **0.453** |
| 150 | 0.731 | 0.170 | 0.450 |

*Table S18.* Ablation Study of the number of Transformer layers on the AudioCaps benchmark.

| number | CIDEr | SPICE | SPIDEr |
|--------|-------|-------|--------|
| 4 | 0.696 | 0.166 | 0.431 |
| 8 | **0.735** | 0.171 | **0.453** |
| 16 | 0.653 | **0.174** | 0.414 |

## C.3. Some Instances of MATS-LLaMA

In Figure S6, Figure S7, Figure S8, Figure S9, and Figure S10, we present examples of MATS-LLaMA and Qwen2-Audio-Instruct (Chu et al., 2024) applied to tasks including complex QA, audio classification, general audio captioning, simple QA, and music captioning. In the AIR-Bench Chat and MusicCaps benchmark examples, Qwen2-Audio-Instruct exhibits significant hallucination issues, whereas MATS-LLaMA, while only partially covering the answers, does not generate incorrect or nonsensical responses. For audio event classification tasks (FSD50K, BJO, and US8K), Qwen2-Audio-Instruct struggles to generate specific responses, often providing only broad and general answers. For audio captioning examples (AudioCaps and Clotho), both Qwen2-Audio-Instruct and MATS-LLaMA can correctly generate appropriate captions. Additionally, in the ClothoAQA example, the correct answer is 'BIRD'. While Qwen2-Audio-Instruct's response ('owl') refers to a bird, the sound does not correspond to that of an owl. These cases highlight MATS-LLaMA's strong audio understanding and reasoning capabilities.

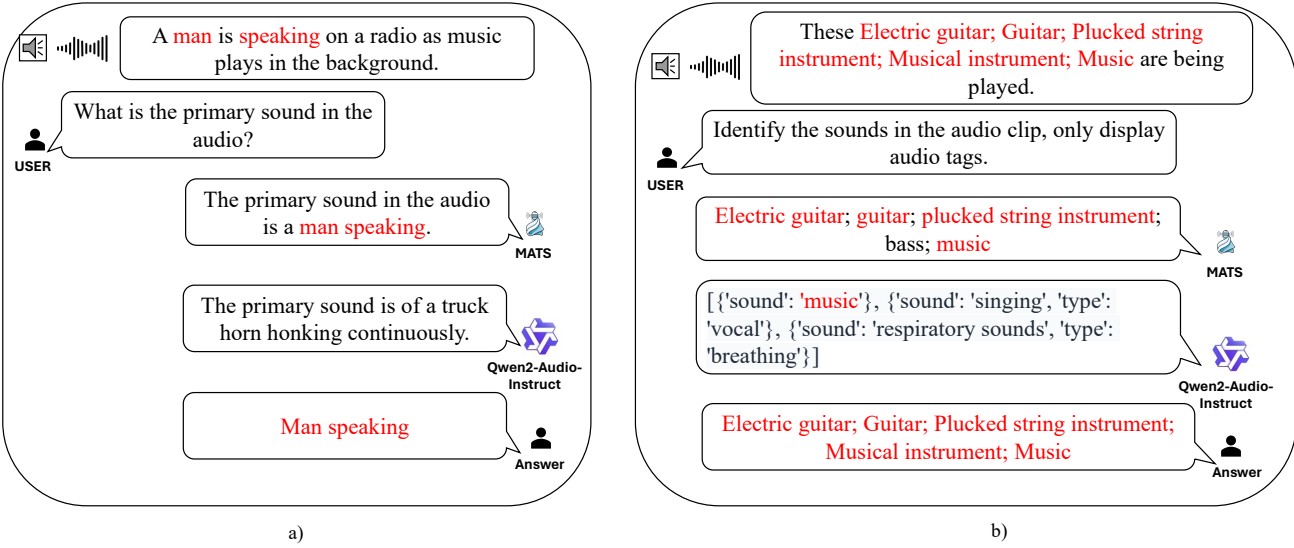

*Figure S6.* The example of different tasks. a) Complex audio QA tasks on AIR-Bench Chat Benchmark. b) Audio Classification tasks on FSD50K.

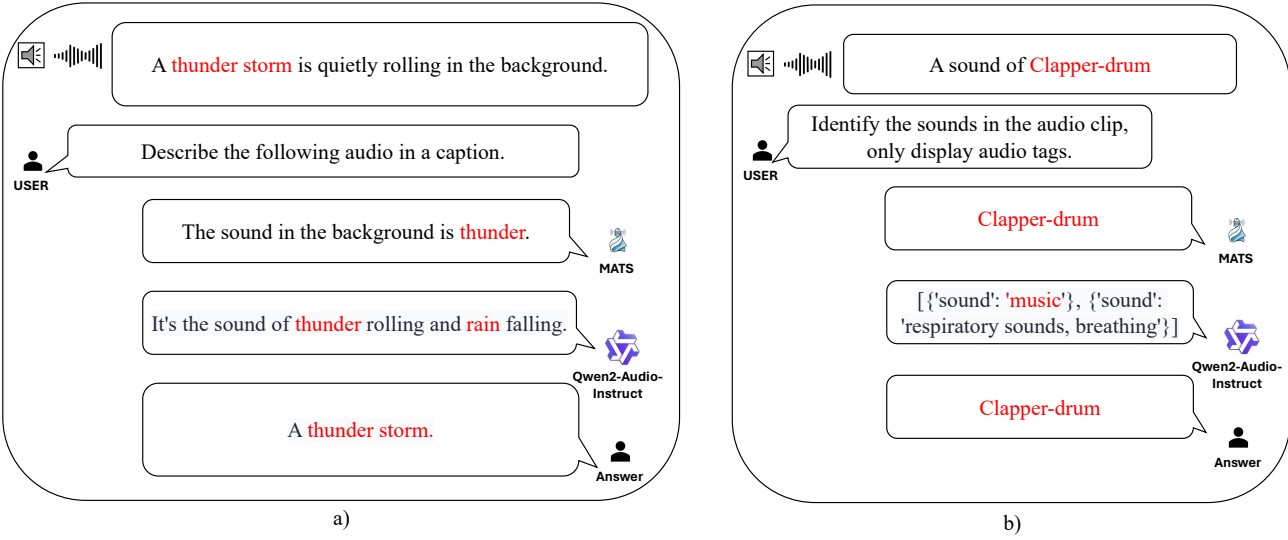

*Figure S7.* The example of different tasks. a) Complex audio QA tasks on AIR-Bench Chat Benchmark. b) Audio Classification tasks on BJO.

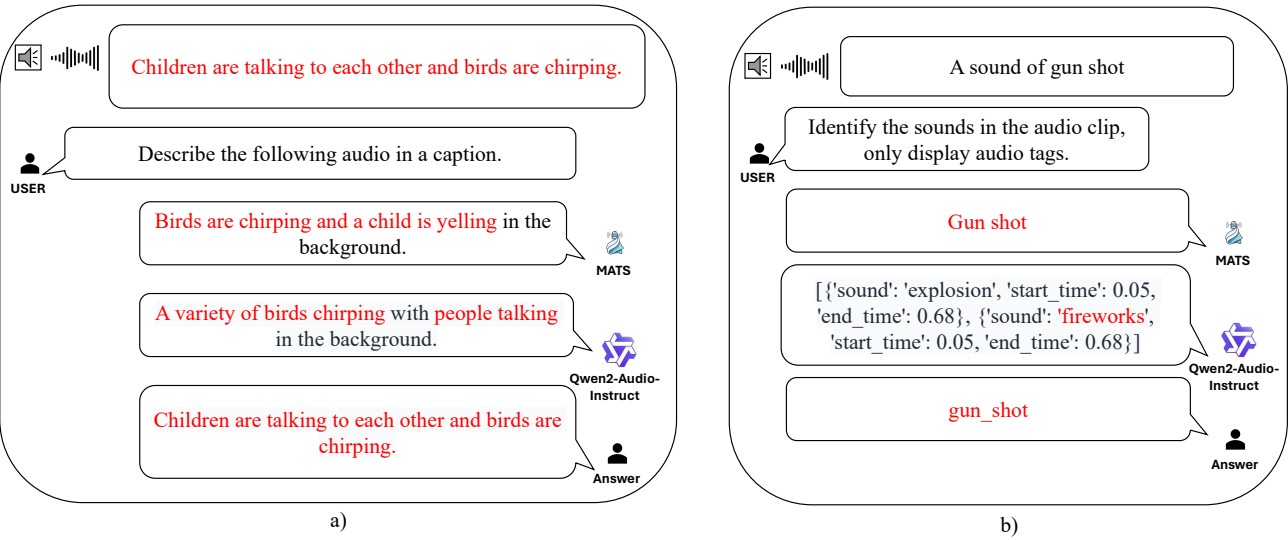

*Figure S8.* The example of different tasks. a) Audio Captioning tasks on Clotho. b) Audio Classification tasks on UrbanSound8K.

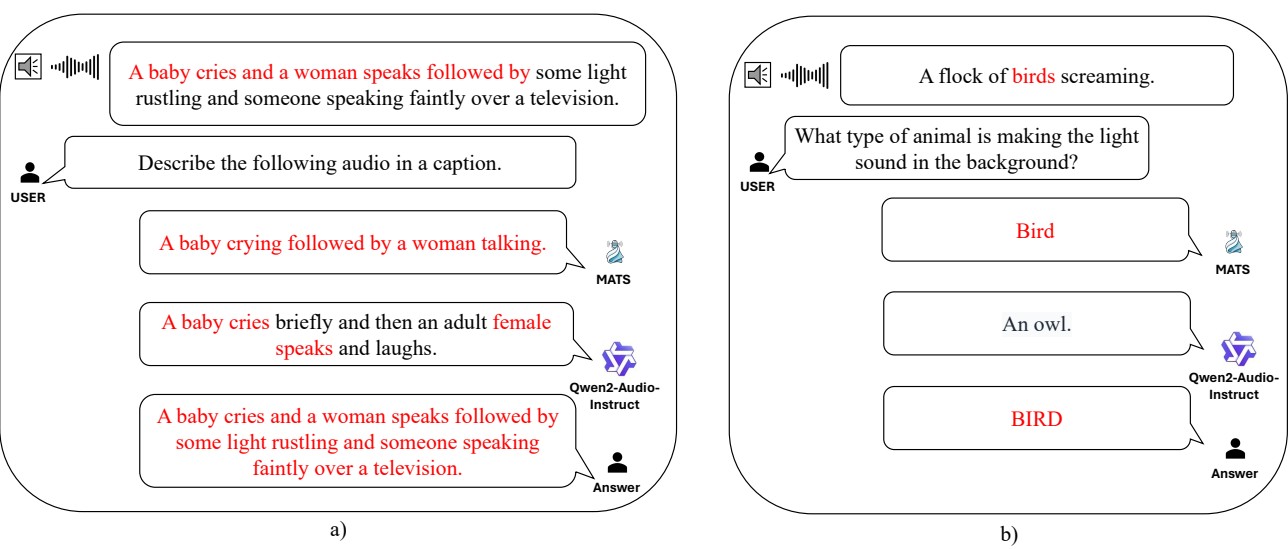

*Figure S9.* The example of different tasks. a) Audio Captioning tasks on AudioCaps. b) Simple audio QA tasks on ClothoAQA.

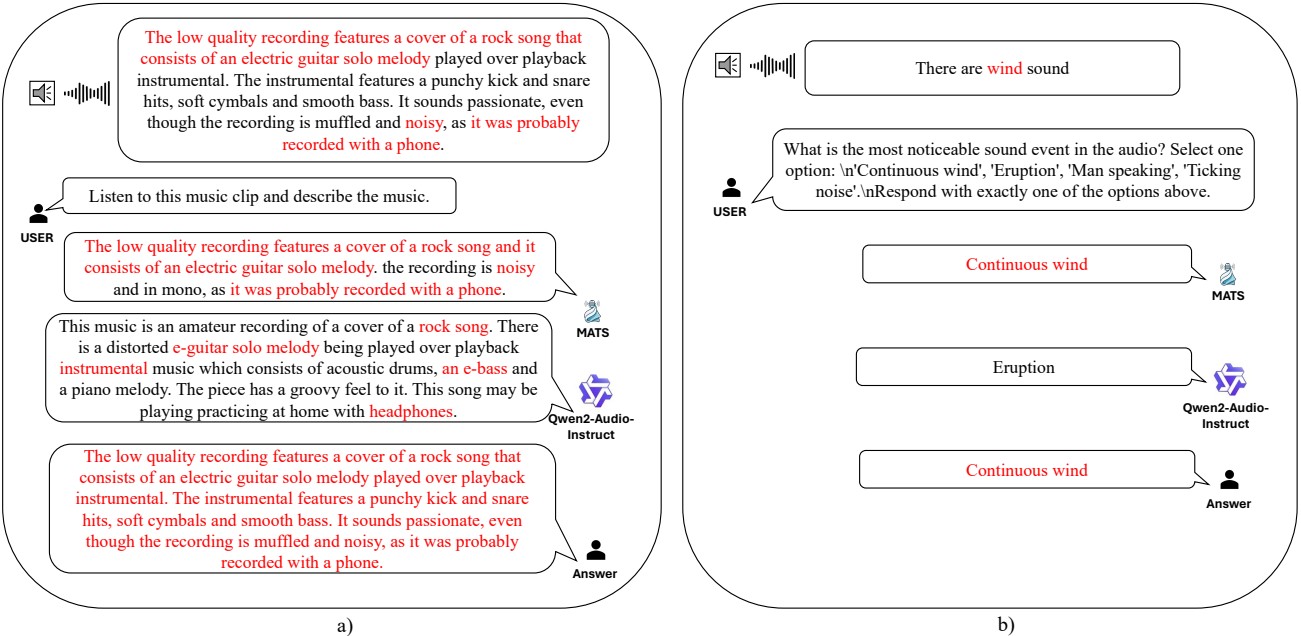

*Figure S10.* The example of different tasks. a) Music Captioning tasks on MusicCaps. b) Complex audio QA tasks on MMAU Benchmark.

