# OpenReview forum: "MATS: An Audio Language Model under Text-only Supervision"
_ICML.cc/2025/Conference — ICML 2025 poster_

### Official Review · Reviewer_LCoU · 2025-03-06

**Overall Recommendation:** 3

**Summary:**

The authors propose to use pre-trained audio-text contrastive models such as CLAP to achieve text only supervision, with a strongly-related noisy text with audio mechanism to introduce robustness.

**Claims And Evidence:**

The authors compare proposed methods to several other audio large language models and show that proposed methods can perform similarly to those trained with the need of audio data during training phase. This provide evidence that proposed method works to some degree.

**Essential References Not Discussed:**

NA

**Experimental Designs Or Analyses:**

The authors also provide MATS-Audio to show the modality gap of approximate with CLAP models. This is a good additional contribution to the community.

**Methods And Evaluation Criteria:**

The authors leverage various and diverse audio tasks and datasets for evaluation, this shows more robust and generalizable results.

**Other Comments Or Suggestions:**

NA

**Other Strengths And Weaknesses:**

The comparisons with various open-ended audio tasks such as AIR-Bench and MMAU provide thorough understanding of the current landscape.

**Questions For Authors:**

NA

**Relation To Broader Scientific Literature:**

Multimodal large language model is a current popular research, this work explore the modality gap in multimodal encoders and how it affects when trained with large language model, which is a good contribution to the community.

**Theoretical Claims:**

The authors provide a theoretical analysis on generalization and take the modality discrepancy into consideration, this helps to provide different perspectives to the problem other than based on heuristics.

---

> ### Author Rebuttal · Authors · 2025-03-31
>
> ### We sincerely appreciate your time and effort in reviewing our manuscript. Your positive evaluation is highly encouraging. Thank you for your valuable feedback.

---

### Official Review · Reviewer_pS8x · 2025-03-07

**Overall Recommendation:** 3

**Summary:**

This paper proposes a text-only supervision method that closes the gap between the text embedding space and the audio embedding space via a mechanism called santa.

## Update after rebuttal
I deeply appreciate authors providing additional results. It resolves my other concerns except for this one: "connection between the bound derived and the proposed "santa" method is not clear". I have read the rebuttal about this but still feel not that directly related and felt the derivation a bit disjoint with the main theme of the paper.

Therefore, I decided to maintain my score as it is.

**Claims And Evidence:**

Yes

**Essential References Not Discussed:**

No

**Experimental Designs Or Analyses:**

Yes. The experiments are sound to me.

**Methods And Evaluation Criteria:**

Yes

**Other Comments Or Suggestions:**

Lines 165-177 on left column of page 4 seem to be a repetition of stuff on page 3
line 264: identity -> identify

**Other Strengths And Weaknesses:**

Other Weaknesses:
1. I found the connection between the bound derived and the proposed "santa" method not very clear. It reads to me as if the authors used a lot of maths to prove a bound, only to conclude that we need to bridge the two spaces closer and propose a method that does not directly use the disc_L1 metric. Please clarify.

**Questions For Authors:**

1. Why do you need 5M text samples? Is the description space that large?

**Relation To Broader Scientific Literature:**

This paper has compared to a wide range of existing work that can perform audio understanding.

**Theoretical Claims:**

Yes. The proof of theorem 3.1 is convincing to me provided that the assumption of P_A(y) = P_T(y) is true. This might limit the application scenario to more general tasks (e.g. speech) but is a reasonable assumption for the experiments they conduct.

---

> ### Author Rebuttal · Authors · 2025-03-31
>
> ### **Q1: The proposed Santa does not directly use the disc_L1 metric.**
>
> 1. In our design, we only have access to text-only data during training,  making it impractical to directly use the disc_L1 metric to reduce the modality gap. Instead, as shown in Figure 3 of main paper, our Santa achieves a similar effect in effectively reducing the distance between audio and text embeddings.
> 2. Further, Table 1 presents relevant statistical information. Specifically, we randomly select 350 samples from `AudioAIA` dataset and generate their language embeddings $Z_T$ and audio embeddings $Z_A$ using the CLAP encoder. Santa is then applied to the audio embeddings, denoted as $f_{\mathrm{Santa}}(Z_A)$.
> Next, we calculate the L1 distance between the prototype of audio embeddings $Z_A/f_{\mathrm{Santa}}(Z_A)$ and language embeddings $Z_T$. As shown in Table 1, Santa effectively reduces the distance between audio and text embeddings, validating its effectiveness in bridging the two spaces closer.
>
>
> | Method      | L1 distance |
> | ----------- | ----------- |
> | Origin CLAP | 18.35       |
> | Santa       | 10.48       |
>
> _Table 1: The statistics on the modality gap within CLAP. **Note:** We randomly select 350 samples to calculate the L1 distance._
>
>
>
> ### **Q2: Lines 165-177 on the left column of page 4 seem to be a repetition of stuff on page 3 line 264: identity -> identify**
> Thanks for your reminder. We will update it in the next version.
>
> ### **Q3: Why do you need 5M text samples? Is the description space that large?**
>
> 1. Using 5M text samples aims to improve the generalization of MATS. With approximately 7B parameters, MATS-LLaMA requires a substantial amount of training data to effectively scale its capacity.
> As shown in Table 2, most existing LALMs of similar size (around 7B parameters) are trained on over 5M audio-text pairs.
>
> 2. Furthermore, we conducted an ablation study to assess the performance of MATS-LLaMA with different training dataset sizes. As shown in Table 3,  the model's performance improves as the dataset size increases, especially in open-ended scenarios. This result validates the necessity of using 5M text samples.
>
> | Model          | Capacity | #Sample |
> | -------------- | -------- | ------- |
> | Audio Flamingo | 2.2B     | 5.9M    |
> | GAMA           | 7B       | 8.7M    |
> | LTU            | 7B       | 5.6M    |
> | LTU-AS         | 7B       | 9.6M    |
> | SALMONN        | 7B       | 5M      |
> | MATS-LLaMA     | 7B       | 5M      |
>
> _Table 2: The number of train dataset from current LALMs_
>
> | Ratio (%) | AudioCaps (CIDEr) | AIRBench-Sound (GPT-4) | MusicCaps (ROUGH_L) | ESC-50 (ACC) |
> | --------- | ----------------- | ---------------------- | ------------------- | ------------ |
> | 50%        | 0.697             | 6.25                   | 16.4                | 0.87         |
> | 75%        | 0.705             | 6.30                   | 17.7                | 0.88         |
> | 100%       | **0.735**         | **6.43**               | **18.7**            | **0.88**     |
>
> _Table 3: Performance of MATS-LLaMA under different training data ratios_

---

### Official Review · Reviewer_72QC · 2025-03-09

**Overall Recommendation:** 3

**Summary:**

This paper proposes MATS, an audio-language multimodal large language model (LALM) that is trained solely on text data while achieving strong performance on various audio comprehension tasks. Unlike conventional LALMs, which require a large corpus of audio-language pairs, MATS leverages CLAP (Contrastive Language-Audio Pretraining) to align audio and language modalities without audio supervision.

During training, MATS only uses textual data, where CLAP's language encoder extracts text embeddings, which are further processed using a Transformer-based mapper before being fed into the LLM. To mitigate the modality gap between CLAP’s audio and text embeddings, a Gaussian noise injection strategy is applied to text embeddings during training.

At inference time, audio inputs are encoded using CLAP’s audio encoder, and the Santa mechanism is introduced to bridge the modality gap. Santa retrieves semantically related caption embeddings from a clustered database and balances them with the input audio embedding. The final input to the LLM consists of both the audio embedding and Santa's retrieved text embedding, effectively improving generalization.

Extensive zero-shot evaluations demonstrate that MATS achieves performance comparable to state-of-the-art audio-supervised models on multiple benchmarks, including audio classification, captioning, and open-ended question answering. Notably, MATS surpasses SALMONN and Qwen-Audio-Chat on the MMAU benchmark while being trained only on text data, showcasing its ability to learn audio semantics without direct audio supervision.

**Claims And Evidence:**

The paper presents experimental results comparing MATS with audio-supervised models, demonstrating that the proposed text-only training method achieves comparable performance. Additionally, the authors claim that the Santa mechanism effectively mitigates the modality gap and outperforms previous text-only audio LLM approaches (as shown in Table 2 and Table 4).

However, I have concerns regarding the justification of the latter claim. From Table 2, MATS appears to perform similarly to previous text-supervised models, and these models are not compared in tasks beyond audio captioning. While I acknowledge that previous text-supervised models primarily focus on captioning, a stronger justification for Santa's superiority is needed. Specifically, since Santa is the key architectural difference from previous text-supervised approaches, a more thorough comparison would be beneficial. This could be done by training an ablated system that replaces Santa with mechanisms proposed in prior works, evaluating it on the broader set of tasks used in this paper. Such an experiment would provide clearer evidence of Santa’s advantage over prior approaches.

Additionally, I am unclear about the discrepancy between the first column of Table 4 and DRCap. What is the exact difference between the mechanism in the first column of Table 4 and DRCap? The performance gap between the first column of Table 4 and DRCap in Table 2 is quite large, and it would be helpful to clarify why this occurs.

**Essential References Not Discussed:**

Not that I know of.

**Experimental Designs Or Analyses:**

Please see "Claims And Evidence" section.

**Methods And Evaluation Criteria:**

Please see "Claims And Evidence" section.

**Other Comments Or Suggestions:**

I would recommend the authors improve Figure 2 to better highlight the correspondence between the upper-right "Modal-Transfer Method" block and the rest of the figure. Currently, there is no clear segmentation between the Santa mechanism and the noise injection component, making it difficult to distinguish these parts. Additionally, clarifying the connection between the label "modal-transfer method" and the Santa/noise injection block would improve the figure's readability.

**Other Strengths And Weaknesses:**

N/A

**Questions For Authors:**

1. In line 252:
   > "However, due to the limited representational power of individual language embedding, this strategy is prone to retrieving the texts with insufficient semantic relevance, thereby affecting the effectiveness of audio-language modality alignment."

   Could you further elaborate or provide evidence explaining **why** individual language embeddings from CLAP have limited representational power? What factors lead to insufficient semantic relevance in this context?

2. In line 405, the authors state that the variance is a hyperparameter and searched for the optimal value. However, in line 169, the author introduces variance as determined by calculating the infinity norm between audio and language embeddings over a set of 30 randomly selected samples. How exactly is noise variance determined?

**Relation To Broader Scientific Literature:**

The key contribution of this paper is proposing a method to train a text-only supervised audio LLM that generalizes to a broader range of audio-related tasks by constructing a more extensive dataset, as well as introducing an improved inference-time mechanism (Santa) to reduce the modality gap between audio and text embeddings. The text-only training framework leverages the recent CLAP model (Elizalde et al., 2023), building upon prior works such as Pengi (Deshmukh et al., 2023) and LTU (Gong et al., 2024). The Santa mechanism further enhances previous memory-based or noise-based methods (e.g., DRCap by Li et al., 2024; NoAudioCaptioning by Deshmukh et al., 2024) by integrating clustering and weighted embedding retrieval, explicitly addressing limitations in existing methods regarding the preservation of audio semantic information.

**Theoretical Claims:**

I read through the theoretical claims in 3.3. I think it is correct but I am not very certain.

---

> ### Author Rebuttal · Authors · 2025-03-31
>
> ### **W1: Training an ablated system that replaces Santa with mechanisms of prior works (PromptAAC and DRCap)**.
>
> Following your suggestion, we replace Santa with the modality-gap reduction mechanism of PromptAAC and DRCap, referred to MATS-PromptAAC and MATS-DRCap. As shown in Table 1, Santa achieves the best performance on both closed-ended and open-ended tasks, which validate that Santa outperforms previous text-only methods.
>
> - DRCap enhances audio captioning performance by leveraging a benchmark-specific memory bank, fully mapping the audio embedding to weighted language embeddings. But DRCap discards the original audio embedding, making its performance heavily dependent on the relevance between the memory bank and the test benchmark.
>   However, in **multi-task setting**, the memory bank is no longer tailored to a single benchmark but instead aggregates information from multiple benchmarks. As a result, the mapping process may introduce unintended noise, projecting the audio embedding into a less relevant textual embedding space, leading to performance drop.
>
> - PromptAAC adopts an augmentation-based approach that involves injecting noise and substituting similar language inputs. It retrieves audio events by matching audio embeddings with language embeddings derived from 527 predefined audio labels in AudioSet. However, the limited variety of audio events restricts the diversity of the retrieved information, resulting in inferior performance compared to Santa.
>
>
> | Benchmark| ESC-50 (ACC) | AudioCaps (CIDEr) | AIRBench-Sound (GPT-4) | AIRBench-Music (GPT-4) |
> | - | - | - | - | - |
> | MATS-PromptAAC| 0.77| 0.593| 6.07| 5.28 |
> | MATS-DRCap | 0.84 | 0.619 | 5.83 | 5.29 |
> | **MATS-LLaMA (Ours)** | **0.88** | **0.735** | **6.43** | **5.76** |
>
> _Table 1: Comparison results on CLS, CAP, and AQA benchmarks._
>
> ### **W2: What is the difference between the mechanism in the first column of Table 4 and DRCap?**
>
> - DRCap introduces the Retrieval-Augmented Generation (RAG) and Projection-Based Decoding (PD) strategies. In Table 4 of main paper, we only use PD strategy (denoted as **Memory-based**).
> - We further report the performance of DRCap (RAG+PD), where we replace Santa with DRCap in our framework. As shown in Table 2, it still underperforms the performance of DRCap (reported in original paper) in single-task setting .
> - This is because DRcap fully discards original audio embedding during inference, making its performance heavily dependent on the relevance between the memory bank and test benchmark.
>     However, in multi-task setting, the memory bank is no longer tailored to a specific benchmark but instead integrates information from multiple benchmarks. This broader integration can introduce unintended noise during the mapping process, projecting the audio embedding into a less relevant textual embedding space. As a result, MATS-DRCap, trained in a multi-task setting, experiences a performance drop compared to DRCap.
>
> | Method| CIDEr | SPICE | SPIDEr |
> |-|-|-|-|
> | Memory-based (Only PD) | 0.234 | 0.094 | 0.164  |
> | MATS-DRCap (PD+RAG) | 0.619 | 0.175 | 0.397 |
> |DRCap (single-task setting reported in original paper) | 0.718 | **0.186** | 0.452
> | **MATS-LLaMA** | **0.735** | 0.171 | **0.453** |
>
> _Table 2: Ablation Study on AudioCaps._
>
> ### **W3: Improve Figure 2.**
>
> Thanks for your suggestion. We will update the figure in next version to better illustrate the "Modal-Transfer module" and its connection to noise injection/Santa.
>
> ### **Q1: Could you elaborate or provide evidence explaining why individual CLAP text embeddings have limited representational power? What factors lead to insufficient semantic relevance?**
>
> 1. To validate it, we perform a retrieval task between CLAP audio embeddings and CLAP text embeddings on Clotho test set. Specifically, we compare the error rates of two strategies: top-K retrieval; K-means clustering followed by top-K retrieval. As shown in Table 3, the K-means method achieves a lower error rate in capturing semantically relevant captions, effectively mitigating the impact of irrelevant textual information caused by the limited representational capacity of individual language embeddings.
>
> 2. It may be attributed to the  CLAP text encoder compressing textual information into a 1024-dimensional embedding space. Such aggressive dimensionality reduction leads to a loss of fine-grained semantic details, resulting in insufficient representational capacity of individual text embedding.
>
> |Method|Error Rate@5|
> | - | - |
> |K-means-based | 18.3% |
> |TopK | 23.3% |
>
> _Table 3: Retrieving Error Rate@5 on the Clotho Test Set_
>
> ### **Q2: How exactly is noise variance determined?**
>
> The variance is treated as a hyperparameter. As suggested by [1], the optimal value roughly aligns with the used strategy (calculating infinity norm between audio and text embeddings over 30 randomly selected samples), also shown in Figure 4 of the main paper.
>
> [1] Training audio captioning models without audio.

---

> > ### Comment · Reviewer_72QC · 2025-04-04
> >
> > Thank you for clarifying my questions and addressing my concerns. I have increased my score to 3 accordingly. It would be great to see those clarifications included in the updated paper.

---

> > > ### Author Response · Authors · 2025-04-05
> > >
> > > ### We sincerely appreciate your thoughtful and constructive feedback. We are especially grateful for your recognition of our work. Your feedbacks were valuable in helping us improve the quality and clarity of the paper. And we will incorporate the clarifications and improvements into the revised version of the paper. Thank you again for your time and efforts in reviewing our submission.

---

### Decision · Program_Chairs · 2025-05-01

**Decision:**

Accept (poster)

**Comment:**

This paper proposes MATS, a text-only trained audio-language model that achieves performance comparable to audio-supervised LALMs.
It introduces SANTA (Strongly-related noisy text with audio), a novel method that bridges the modality gap between CLAP’s audio and text embeddings during inference. Reviewers agree this is a solid contribution

MATS represents a clear step forward in training multimodal models under modality-limited supervision, which is both a practical and theoretically interesting direction. The paper introduces a novel method (SANTA), demonstrates competitive empirical results, and shows thoughtful evaluation across multiple dimensions.